# Evaluation of Nutrient Intake and Food Consumption among Dutch Toddlers

**DOI:** 10.3390/nu13051531

**Published:** 2021-05-01

**Authors:** Elly Steenbergen, Anne Krijger, Janneke Verkaik-Kloosterman, Liset E. M. Elstgeest, Sovianne ter Borg, Koen F. M. Joosten, Caroline T. M. van Rossum

**Affiliations:** 1National Institute for Public Health and the Environment, 3720 BA Bilthoven, The Netherlands; elly.steenbergen@rivm.nl (E.S.); janneke.verkaik@rivm.nl (J.V.-K.); sovianne.ter.borg@rivm.nl (S.t.B.); 2Erasmus MC, University Medical Center Rotterdam, 3015 GD Rotterdam, The Netherlands; j.j.a.krijger@erasmusmc.nl (A.K.); l.elstgeest@erasmusmc.nl (L.E.M.E.); k.joosten@erasmusmc.nl (K.F.M.J.)

**Keywords:** dietary intake, macronutrients, micronutrients, food groups, dietary guidelines, young children

## Abstract

Improving dietary habits at a young age could prevent adverse health outcomes. The aim was to gain insight into the adequacy of the dietary intake of Dutch toddlers, which may provide valuable information for preventive measures. Data obtained from the Dutch National Food Consumption Survey 2012–2016 were used, which included 672 children aged one to three years. Habitual intakes of nutrients were evaluated according to recommendations set by the Dutch Health Council. Specific food groups were evaluated according to the Dutch food-based dietary guidelines. For most nutrients, intakes were estimated to be adequate. High intakes were found for saturated fatty acids, retinol, iodine, copper, zinc, and sodium. No statement could be provided on the adequacy of intakes of alpha-linoleic acids, N-3 fish fatty acids, fiber, and iron. 74% of the toddlers used dietary supplements, and 59% used vitamin D supplements specifically. Total median intakes of vegetables, bread, and milk products were sufficient. Consumption of bread, potatoes and cereals, milk products, fats, and drinks consisted largely of unhealthy products. Consumption of unfavorable products may have been the cause of the observed high and low intakes of several nutrients. Shifting towards a healthier diet that is more in line with the guidelines may positively affect the dietary intake of Dutch toddlers and prevent negative health impacts, also later in life.

## 1. Introduction

A healthy diet, characterized by an adequate, safe, and balanced nutritional intake, is pivotal in preserving and promoting overall health throughout the life course [1]. Early childhood is a period of rapid growth and development, and therefore, a time of great opportunity, yet also vulnerability. Hence, nutrition during early life is of special importance and increasingly recognized for its long-term implications [2]. Undernutrition during childhood, defined as insufficient intakes of energy or nutrients, has been linked to short-term consequences, such as impaired growth and development as well as higher infection and mortality risk [3]. In addition, undernutrition is also related to later life health consequences, such as the increased risk of diabetes and hypertension. In addition, an inadequate diet might also have sociodemographic consequences in the long-term, such as lower education level and lower income, due to poorly developed cognitive function [4]. Overnutrition comprises the excess and insufficiency of dietary intake along with overweight and obesity. Childhood obesity is associated with various comorbidities, including childhood manifestations of cardiovascular disease, obstructive sleep apnea, non-alcoholic fatty liver disease, and psychosocial problems [5]. Moreover, childhood obesity has been shown to track into adulthood and increases the risk of type 2 diabetes, hypertension, dyslipidemia, and carotid-artery atherosclerosis in those children with persisting obesity [6,7]. Diet-related health consequences are a major threat in public health in Europe as well as worldwide [8].

Although dietary habits established during childhood likely persist into adulthood [9], diet is considered an important modifiable factor [10]. Hence, improving dietary habits at a young age could sustainably prevent adverse health outcomes. In many countries, food-based dietary guidelines are developed to help consumers eat healthily. A healthy diet provides a sufficient intake of nutrients to maintain or improve people’s health. A review on the dietary intake of young children from several European countries has shown potential deficiencies or excess in the intake of nutrients and food groups [11]. However, some of the included studies were conducted more than a decade ago. National food consumption surveys, carried out in several countries, are periodically conducted to provide insight into dietary habits at the population level so that, for example, policymakers and health professionals can implement this in practice by facilitating the shift to more sustainable and safe food for the consumers.

In 2020, a Dutch governmental project was launched on developing a screening tool to assess the nutrition and lifestyle of young children living in the Netherlands, after which measures can be implemented to prevent negative health outcomes. The present study is part of this project and aimed to identify potential nutritional challenges of Dutch children aged one up until three years, which could be considered to be considered in the screening tool. To identify nutritional challenges, the habitual dietary intake, in terms of macronutrients and micronutrients and specific food groups, are described and examined on adequacy by using the most recent food consumption data of the Dutch National Food Consumption Survey (DNFCS 2012–2016) conducted in the general population of the Netherlands [12].

## 2. Materials and Methods

To assess the dietary intake of Dutch toddlers, data of the DNFCS 2012–2016 were used. A detailed methodological description of the DNFCS has been described elsewhere [12].

### 2.1. Data Collection and Study Population

In short, the DNFCS 2012–2016 was a cross-sectional survey carried out among the general Dutch population (1–79 years; n = 4313). Data were collected from November 2012 to January 2017. Participants were recruited from representative consumer panels of Kantar Public, for which the sampling was adjusted for characteristics, such as region of residence, degree of urbanization, educational level, and stratified for age and gender.

General data on background and lifestyle factors of participants were collected from questionnaires. Data on food consumption (intake of foods, drinks, and dietary supplements) were obtained during two nonconsecutive multiple-pass 24 h dietary recalls [13], with an interval of about four weeks, carried out by trained dieticians. The dietary recalls were evenly distributed over the days of the week and seasons.

For the present study, data of 672 children aged one to three years were used. The dietary recalls in this age group were completed by their parent(s) or caregiver(s); the first interview was performed during a home visit (including height and weight measurements by the dietician), and the second one was by telephone. To cover any consumptions at the daycare or elsewhere, the parent(s) or caregiver(s) completed a food diary for their child the day before the interviews took place.

To calculate macronutrient and micronutrient intake, food consumption data were combined with an extended version of the Dutch Food Composition Database (NEVO-online 2016) [14] and the Dutch Supplement Database (NES) dated 1 January 2018 [15]. In addition, the foods were classified into food groups according to the “wheel of five”, which is substantiated by the Dutch food-based dietary guidelines [16]. Within this classification, products were distinguished into products that meet the Dutch food-based dietary guidelines (within the wheel of five) and products for which it is advised not to consume or to limit the consumption (outside the wheel of five). In addition, the wheel of five provides general recommendations on food consumption [17].

### 2.2. Data Analyses

Descriptive statistical analyses of participants’ general characteristics were performed for the study population, unweighted and weighted for sociodemographic properties for which a weighting factor was applied to the participants in the analyses for results to be representative for children aged one to three years in the Netherlands. These general characteristics included characteristics of the participants’ household, supplement use, and fruit and vegetable consumption. Unless otherwise stated, statistical analyses were performed in SAS, version 9.4 [18].

The habitual intake (also referred to as usual intake) distribution of macronutrients, micronutrients, and food groups was estimated from the observed daily intake by correction for the intra-individual (day-to-day) variance, using the Statistical Program to Assess Dietary Exposure (SPADE version 3.2.52 in R, [19]). SPADE analyses were performed age-dependently by gender, using data from all subjects in DNFCS 2012–2016 to predict the model parameters. Results were combined for specific age groups, e.g., children aged one to three years. For most nutrients, the SPADE one-part model was used. Different models were used for folic acid (two-part model) and micronutrients, fiber, and N-3 fish fatty acids (three-part model). If relevant, usual nutrient intakes from food, dietary supplements, and discretionary salt used at the table or during preparation were modeled separately and subsequently combined to total the usual intake (first shrink then add) [20,21]. For iodine and sodium, salt added during preparation or at the table was considered. To estimate the intake from different sources, a multipart model was used. To estimate habitual food consumption, different SPADE models were used for food groups consumed episodically (two-part model) and daily (one-part model). For more details, see the report on the DNFCS 2012–2016 [12].

Results for children aged one to three years are shown in terms of the mean and the distribution of the habitual nutrient intake and food consumption per day (percentiles 5, 25, 50, 75, and 95). 95% confidence intervals were estimated for the mean and the median (50th percentile) using bootstrap analyses.

### 2.3. Evaluation of Intake and Consumption

The habitual intake distributions of macronutrients and micronutrients from food only and from food and dietary supplements, if relevant, were evaluated by comparison with the ad-interim Dutch dietary reference intakes set by the Health Council in 2014 [22]. The evaluation method differed depending on the type of dietary reference value that was available. The estimated average requirement (EAR) of nutrients was used to estimate the proportion of Dutch toddlers with inadequate intake, using the EAR cut-point method [23]. If the proportion was less than 10%, the nutrient intake was considered adequate by a rule of thumb. When the EAR was not available, the adequate intake (AI) was used, which qualitatively evaluates whether a low prevalence of inadequate nutrient intake could be assumed [24]. If the median intake was at or above the AI, the intake seemed adequate. If the median intake were below the AI, no statement could be provided on the risk of inadequacy and further research on the intake is required. The evaluation with an EAR or AI does not indicate whether the intake is adequate or tolerable but only indicates the probability of adequacy.

For vitamin D, the intake evaluation was performed by comparing the intake with the AI, which was set assuming sufficient exposure to sunlight (i.e., 3 µg). It was assumed that two-thirds of the requirement was covered by vitamin D production in the skin by sunlight exposure with light skin types [25]. The AI for vitamin D intake when sunlight exposure is insufficient is 10 µg. For energy, the intake could not be evaluated by the EAR cut-point method, as one of the underlying assumptions (i.e., intake and requirement are not correlated) was not met. For vitamin K_1_, no estimations were made for the intake from food and supplements as no data were available on vitamin K_1_ in the NES database.

The tolerable upper intake levels (UL) for nutrients set by European Food Safety Authority (EFSA) [26] were used to estimate the proportion of Dutch toddlers that may be potentially at risk of adverse effects due to excessive intake of a nutrient. If this proportion (whose intake exceeded the UL) was larger than 2.5%, the nutrient intake was considered high at a population level. Otherwise, the intake was considered tolerable by a rule of thumb.

The habitual consumption distribution of food groups was evaluated by the wheel of five and the Dutch food-based dietary guidelines [17]. Recommendations of intakes of vegetables, fruit, and bread were set in terms of a range. For the intake evaluation, it was assessed per food group whether the median intake was equal to or larger than the recommended intake (or higher than the lower bound of the range) for products within the wheel of five (“in”) and for all products within and outside the wheel of five (“total”). For the food groups, cheese and meat, the guideline was a maximum consumption, and it was assessed whether the median intake was below that recommendation.

## 3. Results

### 3.1. Population Characteristics

The population characteristics of the study population are shown in Table 1. Within the study population, there was an even distribution of boys and girls, of which the majority had a normal BMI. Eight percent of the study population was overweight or obese, and eight percent was (seriously) underweight. The migration background of the children’s parents was mostly Dutch, and most of the parents had finished at least a middle education. Household sizes varied (between two to five persons), of which mostly consisted of four persons. Relatively more households were located in the west, corresponding with the most densely populated area of the Netherlands. From the questionnaires, it was observed that 77% of the toddlers had a daily consumption of fruits and 50% a daily consumption of vegetables. Furthermore, 74% of the toddlers used dietary supplements in general. In total, 59% used vitamin D supplements, and 72% used vitamin D-containing supplements (i.e., vitamin D, a combination of calcium and vitamin D, multivitamins, including minerals, and multivitamins without minerals) in winter and/or during the rest of the year.

### 3.2. Habitual Nutrient Intake

The habitual mean intake and percentiles of the intake distribution of macronutrients and micronutrients are shown in Table 2a,b, respectively.

The intakes of total protein, total fat, polyunsaturated fatty acids, cis-unsaturated fatty acids, trans-fatty acids, linoleic acid, and total carbohydrates met the recommendations of adequate and safe intakes. The total protein intake was adequate as the median protein intake (13.0 En%) was larger than (more than twice) the AI (5.0 En%). Four percent of the toddlers had an intake of saturated fatty acids above the UL. No statement on inadequacy was possible for alpha-linoleic acids and N-3 fish fatty acids (EPA + DHA) as the median intakes were below the AI. The median intake of N-3 fish fatty acids (EPA + DHA) was almost four times slower than the AI. The median fiber intake was below the recommended level.

The intakes of vitamins B_1_, B_2_, B_3_, B_6_, B_12_, C, E, and K_1_, as well as folate equivalents, folic acid, calcium, magnesium, potassium, and selenium, met the recommendations. Under the assumption of sufficient sunlight exposure (i.e., two-thirds of the requirement was covered by vitamin D production in the skin by sunlight exposure with light skin types) for the toddlers, the median vitamin D intake from food and supplements was higher than the AI (as shown in Table 2b); thus, the intake met the recommendation. However, when using the AI for vitamin D intake when sunlight exposure is insufficient (i.e., 10 µg), the median vitamin D intake from food and supplements was below that AI. The intake of retinol from food only and from both food and dietary supplements was considered high as the proportion exceeding the UL was 7.9% and 10.5%, respectively. The median intake of retinol activity equivalents (RAE) from both food only (508 µg) and food and supplements combined (533 µg) was above the AI (300 µg). Therefore, there was a low risk of inadequate intakes. For copper and zinc, the intakes seemed to be adequate according to the AI. However, high intakes of copper and zinc from both food only as from food combined with supplements were observed (for copper 10.2% and 11.5%, and zinc 18.6% and 24.3%, respectively had an intake above the UL). For iodine via food combined with dietary supplements, the intake was considered high for a subgroup of the children (5.1% exceeded the UL). For iron, the median intake from food only (4.6 mg) as well as from food and dietary supplements (4.8 mg) was quite below the AI (8 mg); therefore, no statement on inadequacy could be provided. On the contrary, the median intake of vitamin C and magnesium was twice the AI. Sodium intake was considered high as the proportion exceeding the guideline of 6 g per day was 47.5%. Except for vitamin D, no major differences were observed between the intake via food or via food combined with dietary supplements.

Not reported in tables is the habitual intake of energy. The EAR for the energy intake was 5 MJ per day, and the observed median intake was 5.2 MJ per day. However, the energy intake could not be evaluated with the EAR.

### 3.3. Food Group Consumption

The mean habitual consumption and percentiles of the consumption distribution of food groups mentioned in the wheel of five are shown in Table 3. For each food group, the consumption was compared with recommended consumption levels and evaluated for products that fit the wheel of five (categorized as “in” the wheel of five) and the “total” consumption (in and outside the wheel of five). Evaluation of food groups that do not consist of products that fit the wheel of five (“out”) are not shown in Table 3 as there are no recommended consumption levels for these products. However, it is recommended to limit the consumption of products that do not fit the guidelines.

The total median intakes (thus, of products both in and outside the wheel of five) of vegetables, bread, and milk products were larger than the (lower bound of the) recommended consumption levels. The 95th percentile of the consumption of these food groups equaled to or exceeded the (lower bound of the) recommendations. However, the median intake of products that fit the wheel of five of these food groups remained below the recommended consumption levels. For several food groups, less than 25% of the toddlers consumed following the recommendations (legumes and pulses, nuts, fish, eggs, and fats). For the food groups bread, potatoes and cereals, milk products, fats, and drinks, a large part of the total consumption came from products outside the wheel of five, despite the guidelines to minimize the consumption of sugar-sweetened beverages, to replace refined grains with whole wheat and whole-grain products, and to replace solid fats and butter by liquid fats, margarine and plant-based oils.

Of the food groups, of which all products are categorized outside the wheel of five, the daily consumption was the highest for snacks. It is recommended that toddlers do not consume cheese (0 g per day); however, in practice, they do (median intake is 10 g per day). The median intake of meat was 33 g per day, close to the recommended maximum level of 35 g per day.

## 4. Discussion

In the present study, it was observed that for most nutrients, the estimated habitual intake of Dutch children aged one to three years met the recommendations for adequate and safe intakes. However, there are still opportunities for improvement of the nutrient intake and food consumption of these children.

For toddlers in several other European countries, results similar to those of the present study were found. The intakes of N-3 fatty acids, iron, and vitamin D and the consumption of vegetables were consistently below recommended levels, while intakes of saturated fatty acids, sodium, free sugar, and protein were often higher than recommended levels [11].

Compared to a previous study of the DNFCS among young children, conducted in 2005–2006, similar results were found regarding the consumption of vegetables and fruit and the intakes of fiber, retinol, iron, copper, and zinc [31]. The results refer to children aged two to three years rather than to children aged one to three years as in the present study; however, similar conclusions were drawn. Compared to the previous DNFCS, the folate equivalents intake seemed to be improved [32]. A high intake of copper among young children was also observed [33], for which the main source of copper was cereals and cereal products. In the present study, copper intake is still considered high, and cereal products are still the main source [34]. However, also products, which are not needed for a healthy diet contribute to copper intake. For instance, non-alcoholic beverages (waters excluded) contribute 9.2–11.7% of the copper intake among boys and girls in this age group [34]. In addition, as far as we know, there are no indications of health problems in the Netherlands due to high copper intake reported in the literature; therefore, the copper intake is not considered a dietary nutritional challenge, yet this may be further studied. Vitamin D intake from food and dietary supplements did not meet the AI in the previous study, though it did in the present study. However, in the present study, a lower AI was used, as sufficient sunlight exposure was assumed.

In the present study, 74% of the toddlers used dietary supplements in general, and 59% used vitamin D supplements specifically. The median vitamin D intake from food only was 2.4 µg per day, whereas the median vitamin D intake from food and dietary supplements was 7.6 µg per day. For children in the Netherlands aged up to four years, it is advised to take an additional 10 µg of vitamin D supplements daily [29]. This advice was based on the dietary reference values for adults whose levels below 25 nmol/L were estimated to result in vitamin D deficiency [35]. In 2019, a study on the vitamin D status of Dutch children concluded that one-third of the children were vitamin D deficient in winter, which was likely due to low adherence to the supplementation advice [36]. However, vitamin D deficiency was defined as <50 nmol/L, which is twice the threshold level used by the Dutch Health Council. Nevertheless, more emphasis could be put on compliance with the supplementation advice. Therefore, the intake of vitamin D is a potential nutritional challenge in the dietary habits of Dutch toddlers, depending on the sufficiency of sunlight exposure. In addition, studies on the status of other nutrients, for example, of those of which no statement could be done or of which low intakes were observed in the present study, could be useful in identifying potential nutritional challenges.

For toddlers in the present study, the total protein intake was adequate. However, even the 5th percentile (10 En%) of the protein intake was above the AI (5 En%). Currently, an upper intake level of protein is not yet set. However, a high intake of protein during early childhood is reported to be associated with higher BMI in childhood and a higher risk of obesity in later life [37]. Eight percent of the toddlers in the present study were overweight or obese.

For 50% of the toddlers, it was reported that they ate vegetables every day. The median habitual consumption of vegetable products categorized in the wheel of five was below the recommended level. However, the total consumption of vegetable products (both favorable and unfavorable products categorized in and outside the wheel of five) did meet the recommended consumption level. Toddlers also consumed unfavorable products from several other food groups, especially from bread, potatoes and cereals, milk products, fats, and drinks, which contrasts with the guidelines. The guidelines specifically mention limiting sugar-sweetened beverages, increasing the consumption of whole wheat and whole grain products instead of refined grains, and replacing solid fats and butter with liquid fats, margarine, and plant-based oils. Those products that do not fit the wheel of five are low in fiber or high in unfavorable fats, sugar or salt. The relatively high consumption of unfavorable products may have been the cause for the observed high intake of saturated fatty acids and the median intake of fiber far below the guideline.

As far as we know, no indications of health problems were observed (as it was not examined in the present study) and of insufficient intakes of nutrients. A potential nutritional challenge in the dietary intake of Dutch toddlers is the vitamin D intake, which has been found to be similar for other countries. Therefore, supplementation advice exists for this age group in the Netherlands. However, it remains difficult to assess the adequacy of vitamin D with dietary assessment due to the substantial effect of sunlight exposure. For alpha-linoleic acids, N-3 fish fatty acids, and iron, no statement on adequacy could be provided, though the median intakes were not close to the AI; therefore, these nutrients may be potential nutritional challenges. To gain more in-depth knowledge on potential nutritional challenges and the causal associations between the dietary habits of Dutch toddlers and the impact on their health, further (additional, long-term follow-up) research should be done concerning growth and neuro-development. Insight into the nutrient intake, of which no statement could be done or of which low estimations were observed in the present study, could be provided by additional research, such as on nutritional status. This could be valuable for listing potential nutritional challenges, as was done by studying vitamin D status in Dutch children [36]. In addition, additional analyses within subgroups of this population could potentially provide insight into more class-specific dietary habits related to, for example, age group or socioeconomic status.

There were a few limitations in this study, as in a study involving (self-reporting of) dietary intake, misreporting (underreporting or overreporting) of dietary intake was likely. With self-reporting of dietary intake, misreporting cannot be fully avoided. This is possibly even more the case when the recall day is known. For energy intake, the average level of misreporting than the expected energy intake was estimated as underreported by about ten percent on average, with 2% of the study participants who reported an unlikely low-energy intake [12]. Based on this, the underreporting seems limited. However, bias in the intakes can still not be fully excluded. To estimate the intake of macronutrients and micronutrients, data were combined with the databases NEVO and NES. It is evaluated that the NEVO database is complete though not all products and their declarations are listed and/or available, for which a comparable food product was selected. In the end, the average percentage of missing values for the nutrients presented in this study was only 3% [12]. For the data on supplements, NES uses the nutrient declaration available on the packaging rather than data available through laboratory analyses, which involves average compositions and may lead to overestimation and underestimation of nutrient intake via supplements [38]. In addition, the reference values used for the comparison with the habitual intake of children are ad interim values of the Dutch Health Council, which may be adjusted, as they are working on new reference values for children [39].

For evaluating the intake of food groups, the Dutch food-based dietary guidelines (presented in the wheel of five) were used [16]. However, no compliance with the guidelines does not necessarily mean that the food pattern is inadequate because consumption of various foods and food groups can still lead to adequate intakes of nutrients, as was shown in the present study. The guidelines are set as guidance for individuals rather than for populations. Because the individual requirement is unknown in individual nutritional advice, the recommended daily intake (RDI) is used for guidelines rather than the EAR. The RDI is a value that meets the requirement of 97.5% of the population; thus, for most individuals, it will be more than their individual requirement [29]. For this reason, the EAR cut-point method is usually applied to evaluate the adequacy of intake in populations [23]. Unfortunately, the food-based dietary guidelines are not available in an EAR-like measure. Therefore, in the present study, we made a qualitative comparison of the median consumption of a food group with the guidelines to gain knowledge at a population level rather than assuming that every individual must meet the guidelines.

One of the strengths of the present study was that due to sampling and weighing the results on small deviances on the sociodemographic characteristics. It was possible to obtain results that are representative of the target population. Data were retrieved by using food diaries and repeated 24 h-recalls conform the European guidance for harmonized food consumption data in EU member states by EFSA [40], of which the habitual intake could be estimated and compared with reference values. In addition, of all nutrients from food only as well as from food combined with dietary supplements, the habitual intake was estimated rather than the reported intake on two individual days; therefore, the day-to-day (intraindividual) variation was accounted for, and a better estimate of the proportion with inadequate intakes could be made.

## 5. Conclusions

The dietary intake of Dutch children aged one to three years seems adequate for most nutrients. Vitamin D is a potential nutritional challenge, and several nutrients need to be further looked at for potential nutritional challenges: alpha-linoleic acids, N-3 fish fatty acids, and iron. The dietary pattern of the toddlers consists partially of unfavorable products that may have been the cause of the high intakes of several nutrients, such as sodium and saturated fatty acids, and the low intake of fiber.

Therefore, for young children, shifting to and following a healthy diet, which is (more) in line with the guidelines, may improve the nutrient intake, of which in the present study was found to be low or for which no statement on adequacy could be done. This is important as early-life dietary habits affect health, also later in life. Further research or potential intervention studies on indicators and predictors of a healthy diet for children aged one to three years may be useful to prevent negative health impacts and encourage a healthy life in the future. This knowledge could be incorporated into the screening tool that is being developed for toddlers in The Netherlands.

## Figures and Tables

**Table 1 nutrients-13-01531-t001:** Population characteristics of children aged one to three years in the Netherlands unweighted and weighted for demographic properties (DNFCS 2012–2016; n = 672).

Variable	Categories	Frequency		
		n	% Unweighted	% Weighted
Gender	Male	332	49.4	50.0
	Female	340	50.6	49.9
BMI ^1^	Seriously underweight	18	2.7	3.0
	Underweight	37	5.5	5.4
	Normal weight	563	83.8	83.0
	Overweight	38	5.7	6.4
	Obesity	14	2.1	2.0
	Unknown	2	0.3	0.2
Native country of the parents ^2^	Dutch	622	92.6	92.0
	Western immigrant	17	2.5	2.3
	Non-Western immigrant	33	4.9	5.7
Size of household	Two or three persons	195	29.0	30.0
	Four persons	294	43.8	43.2
	Five or more persons	183	27.2	26.8
Highest education of the parents ^3^	Low	29	4.3	8.0
	Middle	199	29.6	38.0
	High	444	66.1	54.0
Region of household location ^4^	West	303	45.1	47.1
	North	78	11.6	9.8
	East	152	22.6	21.8
	South	139	20.7	21.2
Fruit consumption	Zero to four days per week	59	8.8	9.1
	Five to six days per week	97	14.4	14.4
	Every day	516	76.8	76.5
Vegetable consumption	Zero to four days per week	75	11.2	12.6
	Five to six days per week	257	38.2	37.6
	Every day	340	50.6	49.8
Use of dietary supplements	Yes	504	75.0	74.1
	No	168	25.0	25.9
Use of vitamin D supplements in winter and/or rest of the year	Yes	406	60.4	59.1
	No	266	39.6	40.9
Use of vitamin D containing supplements in winter and/or rest of the year ^5^	Yes	491	73.1	71.9
	No	181	26.9	28.1

^1^ Body mass index (BMI) was calculated per person as the bodyweight divided by the height squared (kg/m^2^). For BMI, age and gender-specific values based on the extended international (IOTF) body mass cut-offs were used [27]. ^2^ Native countries of the parents. Dutch: both parents were born in the Netherlands; Western immigrant: from Europe, United States, Australia; and non-Western immigrant. For Western and non-Western immigrants, at least one parent was born abroad. ^3^ Highest education of the parents. Low: primary education, lower vocational education, advanced elementary education; middle: intermediate vocational education, higher secondary education; and high: higher vocational education and university. ^4^ Region of household location was based on Nielsen CBS division and included the three largest cities Amsterdam, Rotterdam, and The Hague. ^5^ Supplements containing vitamin D: vitamin D only, a combination of calcium and vitamin D, multivitamins, including minerals, and multivitamins without minerals.

**Table 2 nutrients-13-01531-t002:** (**a**)**.** The distribution of habitual macronutrient intake (per day) from food only (“f”) and, if relevant, from food and dietary supplements (“f + s”) by Dutch children aged one to three years (DNFCS 2012–2016, n = 672, weighted for demographic characteristics, season, and day of the week). (**b**) The distribution of habitual micronutrient intake (per day) from food only (“f”) and, if relevant, from food and dietary supplements (“f + s”) by Dutch children aged one to three years (DNFCS 2012–2016, n = 672, weighted for demographic characteristics, season, and day of the week).

**(a)**														
**Macronutrient**	**Source**	**Mean (95% CI)**	**P5**	**P25**	**P50 (95% CI)**	**P75**	**P95**	**EAR**	**% < EAR**	**AI**	**P50 ≥ AI?**	**UL**	**% > UL**	**Evaluation ***
Protein (g/kg)	f	3.1 (3.1–3.2)	1.9	2.5	3.0 (3.0–3.0)	3.6	4.8	0.7	0					EAR: adequate intake
Total protein (g)	f	41 (41–42)	25	34	40 (40–41)	48	61	11	0					EAR: adequate intake
Total protein (En%)	f	13.2 (13.1–13.2)	9.8	11.6	13.0 (13.0–13.1)	14.6	17.1			5	Yes	20	0.5	AI: seems adequate; UL: tolerable intake
Total fat (En%)	f	29.4 (29.3–29.5)	22.1	26.4	29.4 (29.3–29.5)	32.5	36.9			25	Yes	40	1.1	AI: seems adequate; UL: tolerable intake
Saturated fatty acids (En%)	f	11.0 (11.0–11.1)	7.6	9.5	10.9 (10.9–11.0)	12.5	14.8					15	4.2	UL: high intake
Polyunsaturated fatty acids (En%)	f	5.6 (5.6–5.6)	3.6	4.7	5.5 (5.5–5.5)	6.4	8.0					12	0	UL: tolerable intake
Cis-unsaturated fatty acids (En%)	f	15.7 (15.6–15.7)	11.1	13.6	15.5 (15.5–15.6)	17.6	20.7					38	0	UL: tolerable intake
Trans fatty acids (En%)	f	0.3 (0.3–0.3)	0.1	0.2	0.3 (0.3–0.3)	0.3	0.5					1	0	UL: tolerable intake
Linoleic acid (En%)	f	4.6 (4.6–4.7)	2.9	3.8	4.5 (4.5–4.6)	5.4	6.8			2	Yes			AI: seems adequate
Alpha linoleic acid (En%)	f	0.6 (0.6–0.6)	0.4	0.5	0.6 (0.6–0.6)	0.7	0.9			1	No			AI: no statement
N-3 fish fatty acids (EPA + DHA, mg)	f	54 (51–58)	8	20	38 (35–40)	68	158			150	No			AI: no statement
	f + s	57 (52–62)	8	20	38 (36–41)	69	167			150	No			AI: no statement
Total carbohydrates (g)	f	174 (172–175)	100	138	169 (167–171)	204	260	92	3					EAR: adequate intake
Total carbohydrates (En%)	f	54.9 (54.8–55.0)	46.6	51.6	55.0 (54.8–55.1)	58.3	63.1			45	Yes			AI: seems adequate
Fiber (g/MJ)	f	2.4 (2.4–2.5)	1.6	2.1	2.4 (2.4–2.4)	2.8	3.4			2.8 ^1^	No			Guideline: no statement
**(b)**														
**Micronutrient**	**Source**	**Mean (95% CI)**	**P5**	**P25**	**P50 (95% CI)**	**P75**	**P95**		**AI**		**P50 ≥ AI?**	**UL**	**% > UL**	**Evaluation ***
Retinol activity equivalents (RAE, µg) ^1^	f	558 (534–80)	237	371	508 (481–524)	685	1062		300		Yes			AI: seems adequate
	f + s	594 (579–609)	246	387	533 (520–546)	735	1147		300		Yes			AI: seems adequate
Retinol (µg)	f	422 (405–438)	149	256	373 (353–383)	528	874					800	7.0	UL: high intake
	f + s	469 (456–482)	163	282	411 (400–422)	593	969					800	10.5	UL: high intake
Vitamin B_1_ (mg)	f	0.6 (0.6–0.6)	0.4	0.5	0.6 (0.6–0.6)	0.7	1.0		0.3		Yes			AI: seems adequate
	f + s	1.0 (0.6–1.3)	0.4	0.5	0.6 (0.6–0.7)	0.8	1.3		0.3		Yes			AI: seems adequate
Vitamin B_2_ (mg)	f	1.1 (1.0–1.1)	0.6	0.8	1.0 (1.0–1.0)	1.3	1.7		0.5		Yes			AI: seems adequate
	f + s	1.4 (1.0–1.9)	0.6	0.9	1.1 (1.1–1.1)	1.4	2.0		0.5		Yes			AI: seems adequate
Vitamin B_3_ (mg)	f	8.5 (8.5–8.6)	4.8	6.6	8.2 (8.1–8.2)	10.1	13.6		4		Yes			AI: seems adequate
	f + s	9.9 (9.5–10.4)	4.9	6.8	8.6 (8.5–8.8)	11.1	17.0		4		Yes			AI: seems adequate
Vitamin B_6_ (mg)	f	1.0 (1.0–1.0)	0.6	0.8	0.9 (0.9–0.9)	1.1	1.5		0.4		Yes	5	0	AI: seems adequate; UL: tolerable intake
	f + s	1.1 (1.0–1.3)	0.6	0.8	1.0 (1.0–1.0)	1.2	1.8		0.4		Yes	5	0.4	AI: seems adequate; UL: tolerable intake
Folate equivalents (µg) ^2^	f	141 (140–142)	82	111	136 (134–137)	165	219		85		Yes			AI: seems adequate
	f + s	172 (164–179)	88	119	149 (146–152)	193	334		85		Yes			AI: seems adequate
Folic acid (µg)	f	9 (8–10)	0	0	4 (3–5)	13	37					200	0	UL: tolerable intake
	f + s	25 (21–28)	0	1	9 (7–10)	29	106					200	0.7	UL: tolerable intake
Vitamin B_12_ (µg)	f	2.7 (2.6–2.7)	1.2	1.9	2.5 (2.5–2.5)	3.2	4.7		0.7		Yes			AI: seems adequate
	f + s	4.9 (2.3–7.6)	1.4	2.0	2.7 (2.7–2.7)	3.6	5.4		0.7		Yes			AI: seems adequate
Vitamin C (mg)	f	77 (76–78)	29	50	71 (70–72)	96	145		25		Yes			AI: seems adequate
	f + s	96 (75–118)	32	55	79 (77–80)	108	172		25		Yes			AI: seems adequate
Vitamin D ^3^ (µg)	f	2.6 (2.6–2.7)	0.9	1.6	2.4 (2.3–2.4)	3.3	5.3		3		No			AI: no statement
	f + s	8.4 (8.0–8.9)	1.3	3.7	7.6 (6.9–8.3)	11.9	17.5		3		Yes			AI: seems adequate
Vitamin E (mg)	f	7.2 (7.1–7.3)	3.6	5.3	6.8 (6.7–6.9)	8.7	12.0		4		Yes	100	0	AI: seems adequate; UL: tolerable intake
	f + s	8.7 (8.1–9.3)	3.9	5.7	7.4 (7.2–7.5)	9.8	15.8		4		Yes	100	0.3	AI: seems adequate; UL: tolerable intake
Vitamin K_1_ (µg)	f	39.2 (37.3–41.1)	9.5	19.0	30.4 (29.1–31.8)	49.0	98.9		12		Yes			AI: seems adequate
Calcium (mg)	f	700 (696–705)	361	527	671 (665–676)	841	1144		500		Yes			AI: seems adequate
	f + s	720 (711–729)	371	539	686 (679–693)	860	1185		500		Yes			AI: seems adequate
Copper (mg)	f	0.7 (0.7–0.7)	0.5	0.6	0.7 (0.7–0.7)	0.9	1.1		0.3		Yes	1	10.2	AI: seems adequate; UL: high intake
	f + s	0.8 (0.7–0.8)	0.5	0.6	0.7 (0.7–0.7)	0.9	1.1		0.3		Yes	1	11.5	AI: seems adequate; UL: high intake
Iodine (µg)	f	121 (119–125)	70	96	117 (116–121)	143	186		70		Yes	200	2.7	AI: seems adequate UL: high intake
	f + s	127 (125–131)	73	99	122 (119–125)	149	200		70		Yes	200	5.1	AI: seems adequate UL: high intake
Iron (mg)	f	5.8 (5.7–5.8)	3.4	4.6	5.6 (5.5–5.6)	6.7	8.6		8		No			AI: no statement
	f + s	6.2 (6.0–6.4)	3.6	4.8	5.9 (5.8–5.9)	7.1	9.7		8		No			AI: no statement
Magnesium (mg)	f	182 (180–183)	112	148	177 (175–178)	210	267		85		Yes			AI: seems adequate
	f + s	186 (184–188)	115	151	180 (179–182)	215	275		85		Yes			AI: seems adequate
Phosphorus (mg)	f	851 (845–857)	520	691	829 (823–836)	988	1253		470		Yes			AI: seems adequate
	f + s	848 (841–854)	521	690	826 (819–833)	983	1249		470		Yes			AI: seems adequate
Potassium (mg)	f	1840 (1830–1851)	1141	1509	1799 (1787–1811)	2131	2677		1400		Yes			AI: seems adequate
	f + s	1831 (1818–1843)	1131	1495	1790 (1776–1804)	2125	2660		1400		Yes			AI: seems adequate
Selenium (µg)	f	23 (23–24)	13	18	22 (22–22)	27	37		20		Yes	60	0.1	AI: seems adequate; UL: tolerable intake
	f + s	25 (24–25)	14	19	23 (23–24)	29	41		20		Yes	60	0.5	AI: seems adequate; UL: tolerable intake
Sodium (g)	f	3.1 (3.0–3.1)	1.7	2.3	2.9 (2.9–3.0)	3.6	4.8					3 ^4^	47.5	Guideline: high intake
Zinc (mg)	f	5.7 (5.7–5.8)	3.6	4.7	5.6 (5.5–5.6)	6.6	8.4		5		Yes	7	18.6	AI: seems adequate; UL: high intake
	f + s	6.0 (5.9–6.1)	3.7	4.9	5.8 (5.8–5.9)	7.0	8.9		5		Yes	7	24.3	AI: seems adequate; UL: high intake

(**a**) CI = confidence intervals; EAR = estimated average requirement; AI = adequate intake; UL = upper tolerable level. * The habitual intake seemed or was considered to be adequate or tolerable if % < EAR is below 10%, P50 ≥ AI or % > UL is equal to or smaller than 2.5%. ^1^ This is a guideline rather than an AI [28]. (**b**) CI = confidence intervals; EAR = estimated average requirement; AI = adequate intake; UL = upper tolerable level. * The habitual intake seemed or was considered to be adequate or tolerable if % < EAR is below 10%, P50 ≥ AI or % > UL is equal to or smaller than 2.5%. The EAR was not incorporated in the table as there were no values for EAR for the observed micronutrients. ^1^ Calculated as µg retinol + µg β-carotene/12 + µg other carotenoids/24 [29]. ^2^ Calculated using the amount of folate naturally present in foods (in µg) plus 1.7 times the amount of folic acid in enriched foods (in µg) plus 2.0 times the amount of folic acid in dietary supplements (in µg) [14]. ^3^ Assuming that two-thirds of the requirement is covered by vitamin D production in the skin by sunlight exposure with light skin types [25]. ^4^ This is a guideline rather than a UL [30].

**Table 3 nutrients-13-01531-t003:** The distribution of habitual consumption of several food groups (in g/day) by Dutch children aged one to three years, compared to the guidelines of the wheel of five (DNFCS 2012–2016, n = 672, weighted for demographic characteristics, season and day of the week).

Food Group	Wheel of Five *	Mean (95% CI)	P5	P25	P50 (95% CI)	P75	P95	Wheel of Five Recommendation (Min-Max)	P50 ≥ Recommendation?
Vegetables	In	47 (45–49)	14	27	41 (39–43)	60	100	75 (50–100)	No
	Total	56 (54–59)	19	35	51 (48–53)	72	113	75 (50–100)	Yes ^1^
Fruit	In	123 (119–128)	30	74	114 (110–119)	162	249	150	No
	Total	136 (132–140)	36	83	125 (121–130)	179	269	150	No
Bread	In	59 (57–61)	16	37	55 (54–57)	77	115	88 (70–105)	No
	Total	89 (87–91)	41	64	84 (83–87)	109	154	88 (70–105)	Yes ^1^
Potatoes	In	27 (25–29)	7	16	24 (22–26)	35	57	53	No
	Total	38 (36–39)	13	24	34 (32–36)	48	73	53	No
Cereal products	In	4 (3–5)	0	0	0 (0–0)	3	22	38	No
	Total	28 (26–30)	5	13	23 (21–24)	37	69	38	No
Potatoes and cereals ^2^	In	31 (29–33)	7	16	26 (25–28)	41	70	120 (60–120)	No
	Total	63 (61–65)	24	42	59 (57–61)	80	116	120 (60–120)	No
Legumes, pulses	In	2 (1–2)	0	0	0 (0–1)	2	8	4	No
	Total	2 (1–2)	0	0	0 (0–1)	2	8	4	No
Nuts	In	0 (0–1)	0	0	0 (0–0)	0	2	15	No
	Total	4 (4–5)	0	0	2 (2–2)	6	15	15	No
Fish	In	5 (4–6)	0	1	3 (2–4)	6	18	7	No
Eggs	In	6 (5–6)	1	3	5 (4–6)	8	14	11	No
Milk products	In	195 (188–203)	17	91	174 (166–183)	274	446	300	No
	Total	343 (333–354)	96	213	319 (309–329)	448	671	300	Yes
Fats	In	7 (7–8)	2	4	6 (6–7)	10	17	30	No
	Total	14 (14–15)	5	9	13 (13–13)	18	29	30	No
Drinks	In	178 (167–189)	10	54	124 (114–134)	245	527	636	No
	Total	560 (552–568)	198	365	521 (510–527)	713	1057	636	No
Meat	In	8 (8–9)	2	4	7 (6–8)	11	21	35 ^3^	Yes ^4^
	Total	37 (35–38)	13	24	33 (32–35)	46	70	35 ^3^	Yes ^4^
Cheese	In	2 (2–2)	0	0	0 (0–1)	2	10	0 ^3^	No ^4^
	Total	12 (11–13)	2	6	10 (10–11)	16	28	0 ^3^	Yes ^4^
Soups	Out	8 (7–9)	0	0	2 (1–3)	9	38		
Sauces	Out	9 (8–9)	2	4	7 (6–8)	11	22		
Snacks	Out	50 (48–52)	15	29	44 (43–46)	65	105		
Bread toppings	Out	14 (13–14)	2	6	11 (11–12)	18	33		
Other	Out	20 (18–23)	0	1	3 (3–4)	14	91		

* Categorized in the wheel of five (‘in”), outside the wheel of five (“out”), and both in as outside the wheel of five (“total”). ^1^ Within the range of recommendation. ^2^ This food group, including both potatoes and cereals, was included as their products are interchangeable. ^3^ Maximum consumption recommendation. ^4^ In this case, when P50 is equal to or larger than the recommendation, the consumption does not meet the recommended level as it involves a maximum level.

## Data Availability

The data used in this study are available on request from https://www.rivm.nl/en/dutch-national-food-consumption-survey/data-on-request or from the corresponding author (data accessed on 29 July 2020).

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
