# Peer review of "Evaluation of Nutrient Intake and Food Consumption among Dutch Toddlers"

_nutrients, 2021, doi:10.3390/nu13051531_

Round 1

Reviewer 1 Report

Dear authors,

I would like to thank for the opportunity to read your manuscript. From my point of view, it is well-written and you went to great effort in explaining and reasoning your methodic approach. The discussion of the findings is sound and the potential weaknesses are desribed and discussed.

Nonetheless, I have some comments that may help to further strengthen the manuscript and to make it more unique.

1) Introduction: Please describe in more details the mentioned conquences of undernutrition. You name within one sentence health consequences (cardiovascular diseases) as well as potential sociodemographic consequences (lower education and income). From my point of view, both aspects are very important that they should have one sentence each.

2) Please explain - unfortunately, this remained unclear to me - how a project that was conducted between 2012 and 2016 can be part of a new screening tool that was launched in 2020. Are the analyses of the previously collected data part of the current project?

3) In the methods section, you describe that you collected data on sociodemographic characteristics. In the discussion section you describe this a strength. However, I think you can get more out of this. You may consider using the sociodemographic data to analyse food group consumption in more detail. Are there differences between those with a higher socioeconomic status and those with a lower one? Are there differences by immigrant background. All this information would strengthen the manuscript, make it more unique, and it would provide concrete attachment points for health promotion and prevention. Based on these additional analyses, you would be able to give important implications for future research and for prevention. Therefore, I would like to encourage you to perform these additional analyses.

Author Response

We would like to thank the reviewers for reading our manuscript and for providing us with helpful comments. We hope that our modifications in the manuscript and explanations in this letter clarify your comments and questions. 

Reviewer 2 Report

Thank you for the opportunity to review this paper. I have only minor comments. 

Section 2.1 line 86: it sounds as though parents knew when the recalls would occur. This could contribute to under or over reporting and needs to be acknowledged in the limitations (line 368), and also consider how this could have influenced the results. Is diet quality of Dutch toddlers even poorer than these results suggest? 

In that same section - how many children did have data reported by early childhood educators? 

The limitations of assessing usual intake using only two days of dietary intake data need to be acknowledged. Particularly in relation to nutrient intake. For example - paragraph three of the discussion talks about copper intake. This is not a nutrient of concern, for either under or over nutrition, and cannot accurately be assessed with two 24hr recalls anyway.

Link the conclusion back to the purpose (described on line 59) - what are the key dietary components that might be  assessed in a screening tool?

Author Response

(The authors gave the same response as above.)

Round 2

Reviewer 1 Report

Dear authors, Thank you for revising the manuscript. I regret that you do not want to include additional analysis by sex or age. However, I can understand your concerns that the manuscript gets more complex when additional findings are included.